# Looking Towards 2030: Strengthening the Environmental Health in Childhood–Adolescent Cancer Survivor Programs

**DOI:** 10.3390/ijerph20010443

**Published:** 2022-12-27

**Authors:** Laura T. Cabrera-Rivera, Brittney Sweetser, José L. Fuster-Soler, Rebeca Ramis, Fernando A. López-Hernández, Antonio Pérez-Martínez, Juan A. Ortega-García

**Affiliations:** 1European and Latin American Environment, Survival and Childhood Cancer Network (ENSUCHICA), Instituto Murciano de Investigación Sanitaria (IMIB), University of Murcia, 30120 Murcia, Spain; 2Department of Environmental Health, University of Puerto Rico-Medical Sciences Campus, San Juan, PR 00921, USA; 3International Exchange Program for Minority Students, Icahn School of Medicine at Mount Sinai, New York, NY 10029, USA; 4Department of Pediatrics, UC San Diego Health, San Diego, CA 92037, USA; 5Department of Pediatric Hematology & Oncology, Clinical University Hospital Virgen of Arrixaca, 30120 Murcia, Spain; 6Cancer and Environmental Epidemiology Unit, Department of Epidemiology of Chronic Diseases, National Center for Epidemiology, Carlos III Institute of Health, 28029 Madrid, Spain; 7Department of Quantitative Methods and Computing, Technical University of Cartagena, 30202 Cartagena, Spain; 8Pediatric Onco-Hematology Department, La Paz University Hospital, Translational Research in Pediatric Oncology, Hematopoietic Transplantation, and Cell Therapy, IdiPAZ, Autonomous University of Madrid, 28046 Madrid, Spain; 9Paediatric Environmental Health Specialty Unit, Department of Pediatrics, Clinical University Hospital Virgen of Arrixaca, 30120 Murcia, Spain

**Keywords:** environmental health, pediatric cancer, spatiotemporal analysis, carbon footprint

## Abstract

Childhood and adolescent cancer survivors (CACS) are a high-risk population for non-communicable diseases and secondary carcinogenesis. The Environmental and Community Health Program for Longitudinal Follow-up of CACS in the region of Murcia, Spain, is an ongoing pioneering program that constitutes a model for social innovation. This study aims to present the program tools and protocol as a whole, as well as a profile of the incidence, survival, and spatiotemporal distribution of childhood cancer in the region of Murcia, Spain, using 822 sample cases of cancer diagnosed in children under 15 years of age (1998–2020). While the crude incidence rate across that entire period was 149.6 per 1 million, there was an increase over that time in the incidence. The areas with a higher standardized incidence ratio have shifted from the northwest (1998-2003) to the southeast (2016–2020) region. Overall, the ten-year survival rate for all tumor types was 80.1% over the entire period, increasing the five-year survival rate from 76.1 (1998–2003) to 85.5 (2014–2018). CACS living in areas with very poor outdoor air quality had lower survival rates. Furthermore, integrating environmental health into clinical practice could improve knowledge of the etiology and prognosis, as well as the outcomes of CACS. Finally, monitoring individual carbon footprints and creating healthier lifestyles, alongside healthier environments for CACS, could promote wellbeing, environmental awareness, and empowerment in order to attain Sustainable Development Goals for non-communicable diseases in this population.

## 1. Introduction

Globally, the multi-layered processes of preventing and treating cancers during childhood are complex, requiring further attention and research. The trends in the global incidence rate of childhood and adolescent cancer (CAC) are growing; this is becoming the leading cause of death from disease among children in developed countries [1]. In truth, the annual incidence of pediatric cancer in the region of Murcia was 138 (1998–2013) cases per million children (<15 years) [2]. This is at the lower end of the range of incidence in pediatric cancers across European countries, which is 130 to 160 cases per million children under 15 years of age [3]. Among these cases, acute leukemia, lymphomas, and tumors of the central nervous system have the highest incidence [4]. 

Access to specialized centers and new therapies have contributed to increasing survival rates in high-income states. However, inequalities mean that more than 80% of children diagnosed with cancer in high-income countries will live more than five years, and fewer than 40% of children with cancer in low- and middle-income states have the same chance of survival [5,6]. Childhood and adolescent cancer survivors (CACS) are at greater risk of morbidity and mortality during childhood, adolescence, and adulthood, usually due to late relapses, secondary tumors, and increased susceptibility to the early development of non-communicable diseases during adulthood [7,8]. Moreover, the stagnation in survival rates observed in recent years leads us to believe that other external or environmental factors must influence that survival [9,10]. These circumstances have contributed to the development of specific long-term follow-up programs for CACS.

The pollution of ecosystems has also contributed to an increase in the incidence of environmentally related non-communicable diseases such as CAC [11,12,13,14]. The Global Burden of Disease study estimates that 54.8% of global deaths are due to environmental factors, as well as specific lifestyles that are potentially modifiable [12]. Children are especially vulnerable to the deleterious effects of environmental pollutants [13]. Moreover, according to the World Health Organization (WHO), approximately 26% of deaths in children under five years of age can be prevented by addressing environmental risks [14,15]. Environmental risk factors have been associated with the etiology and survival of CAC [11,12,13,14,16]. 

Environmental pollution and climate change play a critical role in cancer risk and surveillance. They increase the amount of carcinogens and disrupt access to healthcare [17]. Since the 19th century, the main cause of climate change has been human activity, mainly due to the burning of fossil fuels such as coal, oil, and gas [18]. Moreover, assessments have shown that a high proportion (60–70%) of total emissions is related to individual decisions and personal lifestyles, including household consumption and transportation [19,20,21]. The total amount of these greenhouse gases generated by our actions is called “the carbon footprint” [22]. According to the WHO, an individual carbon footprint (ICF) is a measure of your activities on the amount of carbon dioxide in air [23]. In Spain, the average carbon footprint is approximately 5.54 tons, a major contributor to climate change [24]. Globally, the average carbon footprint is closer to 4.5 tons [24]. Therefore, the global average carbon footprint per year must drop under 2 tons by 2050 to avoid a 1.5ºC rise in global temperatures [18]. Reducing our carbon footprint requires further efforts be made, not only by governments and private companies, but also by individuals [19]. It is imperative to reduce our ICF to minimize the effects of climate change. Individuals can reduce their ICF by reducing their meat consumption, reducing single-use plastic, and walking or biking instead of using an automobile, among other actions. These actions can also improve our wellbeing through health co-benefits, such as cleaner air, increased physical activity, and more nutritious diets [25].

The development of Pediatric Environmental History (PEHis) and the integration of geographic information system (GIS) have demonstrated its usefulness as a public health tool to formulate hypotheses about the causes of diseases and to help improve socio-health planning and the development of more complex epidemiological studies [26,27,28,29,30,31,32,33,34]. The vast improvements in survival rates achieved in recent decades have changed the paradigm of CAC treatment, going from “cure at any price” in the 60s to “cure at the lowest possible cost” today [35]. Increasingly, the longitudinal follow-up of the CACS must be aimed at achieving the best quality of life in harmony with nature or the environment in which the survivor lives, which will be achieved by controlling the appearance of late effects and knowing the modifiable environmental factors and lifestyles to promote in long-term monitoring programs. 

With this background, and taking as a reference the definition of a survivor of the National Cancer Institute of the United States, which states that a person with cancer is considered a survivor from the moment of diagnosis until the end of their life [36], the Environmental and Community Health Program for Longitudinal Follow-up of CACS in the Murcian region was established in 2004, and includes all patients diagnosed from 1998 to date. This program is focused on offering a holistic follow-up to the survivors and employing an environmental and community-based approach from the moment of diagnosis and throughout their lives. This program focuses on improving the lifestyles and environmental surroundings of the survivor and their family. Networking with primary care professionals and incorporating new skills as well as new professional profiles (environmental health nurse and doctor) are geared to improve the quality of life and wellbeing of CACS. The information collected from this program could help explore possible environmental exposures in the etiology and prognosis of childhood cancer [13]. This study aims to present the program tools and protocol as a whole, as well as a profile of the incidence, survival, and spatiotemporal distribution of childhood cancer in the region of Murcia, Spain.

## 2. Materials and Methods

### 2.1. Data Collection

Murcia is a Mediterranean region in the southeast of Spain with a universal public health system that guarantees the treatment and follow-up of all children diagnosed with cancer. This regional centralized healthcare is in the Pediatric Oncohematology Section and the Pediatric Environmental Health Unit of the University Clinical Hospital of “Virgen de la Arrixaca” (Hospital Clínico Universitario Virgen de la Arrixaca) and facilitates access to almost 100% of the clinical history and professional contact with all children under 18 years of age with cancer and their families. Cases are classified according to the International Classification of Diseases for Oncology (ICD-O-3) [37,38] and ICCC-3 [39]. To avoid misclassifications or duplications, a doctor keeps an annual database log, and families are contacted in person or by phone. In addition, face-to-face interviews and follow-ups with the CACS and their caregivers are carried out by trained health professionals with experience in environmental and community health, pediatric oncology, and risk communication [13,40,41,42]. This study was approved by the University Clinical Hospital of “Virgen de la Arrixaca” Ethics and Research Committee, and informed consent was collected from the parent or legal guardian of the child/adolescent.

### 2.2. Environmental and Community Health Program for Longitudinal Follow-Up of Childhood and Adolescent Cancer Survivors

The CACS Environmental and Community Health Longitudinal Follow-Up Program offers a personalized follow-up to each survivor and has an environmental and community-based approach at diagnosis and throughout the rest of their lives. The program has two key strategies: (a) education and promoting healthier environments and lifestyles for the patient, their family, and the community; and (b) screening and early detection for the adequate intervention of late effects [43]. In order to achieve this, the main tool is the Pediatric Environmental History (PEHis) of CACS.

### 2.3. Pediatric Environmental History of Child and Adolescent Cancer Survivors

Pediatric Environmental History (PEHis) is an innovative, simple, and low-cost instrument that allows an individualized risk assessment to improve an integrative CACS long-term follow-up using an environmental and community approach. Since 2004, PEHis has been performed in all cases of a childhood cancer diagnosis from 1998 in the region of Murcia and included in the “Environment and Pediatric Cancer in the Region of Murcia” (Medio Ambiente y Cáncer Pediátrico en la Región de Murcia or MACAPEMUR) cohort. This individualized risk assessment is included in the standard clinical history of the patient and integrates: (a) treatment-related effects, (b) lifestyle, environmental, and medical history factors, and (c) genetic susceptibility. The PEHis contributes to improving clinical etiological judgment and prognosis and improving education, as well as promoting healthier environments and lifestyles for the survivor, their family, and the community. PEHis was recognized as a good practice by the National Cancer Strategy of the National Health System in 2006 [44]. 

The PEHis of CACS allows clinicians to identify the environmental risk factors and consider clinical records: employing a series of basic and concise questions, including genetic, genealogical, and constitutional aspects. This information is collected as part of the history taken when children are ill, and during survivor supervision visits—reminding the physician to explore possible environmental sources of risk, contaminants, indoor or outdoor exposure, hobbies, occupational exposure, and personal behaviors. The PEHis is part of the CACS standard clinical history. 

The PEHis tool comprises a series of concise and basic questionnaires that integrate: ICF, Green Page, risk assessment, Transition Report, and individualized Long-Term Follow-Up Plan. The PEHis process is shown in Table 1.

The first contact with the program is the Green Page and ICF. It allows personalized environmental health counseling interventions during cancer treatment periods. It is carried out by a pediatrician or nurse with basic training in environmental health. The ICF was recently incorporated into the program and measures the natural resource consumption by humans. The ICF of a cancer survivor is defined as a simple environmental health indicator that aims to reflect all the greenhouse gasses emitted directly or indirectly by the cancer survivor and his family and is estimated in units of carbon dioxide equivalents (CO2e). A low ICF improves wellbeing through health co-benefits, such as cleaner air, increased physical activity, and more healthy, nutritious diets [25]. The ICF has two important limitations: It cannot be calculated exactly because of inadequate knowledge of data about the complex interactions between contributing processes and the need for unified metrics. Although several initiatives have been designed for standardization, the United Nations Carbon Offset Platform [45] is being used. This platform is an international tool that includes household, transport, and lifestyle information of the individual to measure the annual ICF and allows the comparison of results between different regions of the world, with an easy measurement that promotes environmental awareness, empowerment, and healthier lifestyles [45]. 

Furthermore, at the beginning of the process, ideally in the first days after diagnosis, the “Green Page” (Hoja Verde or HverdeGEO) is employed, which is a basic environmental screening [46] including genealogical factors, environmental tobacco-smoke exposure, chemical risk exposure in parents’ occupation, the level of contact with nature, and the ambient air quality. The level of contact refers to the frequency in which activities are conducted with direct contact with nature (e.g., going to neighborhood parks, orchards, mountains, beaches, and forests). The ambient air quality is measured by an air pollution annoyance (or disturbance) scale. This scale relates the perceived risk of families with outdoor air pollution (“How much does air pollution outside their home disturb them if they leave all the windows open”) on an 11-point Likert scale and ranges from 0 (no disturbance at all) to 10 (intolerable disturbance), showing a linear relation and strong correlations with particulate matter less than 10 μm in diameter (PM10) and nitrogen dioxide levels [47]. Furthermore, HverdeGEO collects the postal addresses of CAC cases at three critical time points: during pregnancy, at birth, and at the time of diagnosis. The georeferencing of these three postal addresses, assigning latitude and longitude coordinates to each case, allows the information to be analyzed using Geographic Information System (GIS) tools. These tools create urban incidence maps that identify geographic patterns in small areas (district or census tracts). The temporal and spatial information collected from each case helps to identify spatial or spatiotemporal clusters, or both, ultimately generating an environmental monitoring system [33]. The study of these clusters could also help identify protective and/or risk factors, related to the different tumor types, allowing for more specific subsequent studies [13]. 

The following PEHis process includes the risk assessment for each CACS and their environment, addressing the identification of environmental and constitutional protective and risk factors associated with CAC in general and with each subtype. This is performed “face-to-face” with the parents and the child by trained environmental health personnel, ideally during the first four to six months after diagnosis. This individualized risk assessment lasts between 1 and 3 h. It is a key process for improving the lifestyles and environments of the family survivors during this critical period. In this risk assessment, four major blocks are explored [41,42]. (a) Genealogical–constitutional block, where data are collected on individual and family oncological history up to the third degree of consanguinity; genetic factors associated with infant–juvenile tumors; rare, hereditary, and non-communicable diseases; and causes of death of relatives. (b) Environmental block, where socioeconomic and demographic data; environmental exposures in housing and the neighborhood; occupational and leisure exposures of parents; and consumption or exposure to legal and illegal drugs are explored. (c) Clinical block, where a history of exposure to ionizing radiation in the parents; obstetric history; breastfeeding; and clinical history of the child prior to diagnosis (childbirth, neonatal, radiological history, previous admissions) are collected. (d) Tumor block, where the classification according to ICCC-3; serologies; and cytogenetics is collected. 

As the program progresses, the Transit Sheet or Green Passport (towards primary care) is made when the oncology treatment has been finalized (usually after 2–3 years). It is a personalized report for each CACS with three sections: (a) Clinical, including all the information about the diagnosis, types, and doses of the treatments used, and complications that occurred during the treatment. (b) Late effects, individualized identification of potential late effects associated with treatment, as well as recommended follow-up [48] and description of possible late effects present at the time of completion of the Transit Sheet for each CACS. For the classification of late effects and their grade, the modification of the Common Terminology Criteria of Adverse Events proposed by the St. Jude Lifetime Cohort Study [49] is used. (c) Environment and lifestyles: addresses environmental factors (including lifestyle factors, ICF, contact with nature, etc.); (d) wellbeing and health-related quality of life, and (e) a set of personalized environmental health recommendations and a follow-up plan for surveillance or prevention of late effects. The Green Passport familiarizes the contact of the primary care physician with the survivors, making tracking easy. The goal is to prevent the onset of non-communicable diseases and late effects, and the most frequent therapeutic modality in this section is to achieve changes in the environment and lifestyle. Ultimately, the focus is to motivate and ensure compliance with the therapeutic intervention [40]. More than 750 Green Passports have been issued towards primary care. 

Two to three years after the end of treatment, the process continues with the “Longitudinal Follow-up of Survivors of Childhood-Adolescent Cancer”, which is then continued throughout the survivor’s life. With shared leadership alongside primary care, it guarantees an adequate transition to the primary level of health, carrying out an adequate follow-up based on the individual risks of each CACS that allow an early diagnosis and management of possible late effects, focusing on the promotion and motivation of environmental protective factors, including lifestyles, that reduce the risk of occurrence of such late effects, non-communicable diseases, relapses, and second tumors, and contributing to improve the overall survival and quality of life of CACS [42,43].

### 2.4. Descriptive and Geospatial Study

A descriptive and geospatial study was performed on cases of cancer diagnosed in children under 15 years of age in the region of Murcia, Spain, from 1 January 1998 to 31 December 2020, from the MACAPEMUR cohort, in order to understand the distribution of cases. The present study included those who met the following inclusion criteria: under 15 years of age at diagnosis; a cancer diagnosis that complies with the International Classification of Childhood Cancer (ICCC-3) [38]; diagnosed between 1 January 1998 and 31 December 2020; and a resident of the autonomous community of the region of Murcia at the time of diagnosis. For the analysis, this study only included information recollected from the HverdeGEO and the clinical history of the CACS.

Furthermore, a test of equality of survival distribution was performed for individual and socioeconomic characteristics and environmental exposure of the basic environmental screening (HverdeGEO). The characteristics and environmental exposure were the following: sex, income in EUR/month (<2000, 2000–3500, and >3500), family history of cancer 1st degree, family history of cancer 2nd degree, exposure to pesticides inside the residence, contact with nature (every day, once a week, once a month, only in vacations, never), maternal smoking during pregnancy, paternal smoking during pregnancy, maternal and paternal smoking at diagnosis, and ambient air quality as measured by a Likert air pollution annoyance scale (acceptable, moderate, and very bad) in address at diagnosis. 

### 2.5. Spatial and Statistical Analysis

The descriptive analysis was performed by calculating the means, standard deviations, and frequencies for all variables. Incidence is described in a statistical summary: number of cases, crude rate (CR), and age-adjusted standardized incidence ratio (ASRw) by cancer type and time cohort were calculated with their 95% confidence intervals. In order to estimate overall survival rates, the Kaplan–Meier method was used. Finally, the georeferencing of postal addresses at the time of diagnosis, assigning latitude and longitude coordinates to each case, allowed us to analyze the information using GIS. These GIS tools were used to create urban incidence maps in the different “Health Areas” and “Basic Health Zones” that form the health map of the region of Murcia. “Basic Health Zones” is the most basic geographical health delimitation, each one with its own primary care team with direct access to the population, and “Health Areas” are clearly defined administrative units that brings together a group of primary healthcare centers and professionals under its organizational and functional dependency [50]. All statistical analysis was performed using the IBM SPSS Statistics program for Mac OS X, version 24.0, Armonk, NY, USA: IBM Corp.

## 3. Results

A total of 965 survivors were initially identified, of which 143 were excluded (Figure 1), leaving 822 survivors in the MACAPEMUR cohort.

A total of 822 cases were analyzed. The sample distribution by sex was 448 (54.5%) males and 374 (45.5%) females. The age at the time of this study was <15 years (35.2%), 15-18 years (21.4%), and 18 or more years (43.4%), with a 37-year-old as our oldest follow-up survivor. The mean age at the time of diagnosis was 5.87 years (SD: 4.28), with the most frequent age group being 1–4 years (36.0%), followed by 5–9 years (30.7%), 10–14 years (19.5%), and <1 year (13.9%). The mean duration of follow-up for the CACS in the study was 12.94 years (SD: 5.73).

Table 2 presents the survival distribution according to socioeconomic characteristics and environmental exposure, the overall childhood cancer survival at 2, 5, and 10 years in the region of Murcia from 1998 to 2020, and the log rank (Mantel–Cox) for the overall survival years with a 95th percentile. Females had a higher 2-year survival rate than males (86.3% vs. 82.8%). Additionally, there was a significant difference on the Likert scale of air pollution (<0.05). Figure 2 presents the cumulative survival split as the result of the Likert scale of air pollution (acceptable, moderate, and very bad), in which the group with very bad air pollution had the lowest cumulative survival.

Table 3 presents the childhood cancer distribution using the crude and standardized incidence ratio by cancer type and time cohort. The most common types of cancer were leukemia (29.2%), tumors of the central nervous system (CNST) (23.7%), and lymphomas (10.3%). The crude ratio across all cancer types and time cohorts was 149.6 cases per 1 million children under 15. The SIR between 2010 and 2015 (1.14) was significantly higher from previous years.

The average survival rate from 1998–2018 was 8.62 (CI95% 7.90–8.90), and 82.7% percent of all cases were alive at the time of the study. There were 143 deaths, and the global 5-year survival rate was 81.3% (CI95% 81.1–81.4). Table 4 demonstrates that the 5-year survival rate across all tumor types increased over time and was 76.1% (CI95% 75.7–76.2) from 1998–2003, 80.4% (CI95% 80.1–80.6) from 2004–2008, 82.0% (CI95% 81.8–82.1) from 2009–2013, and 85.5% (CI95% 85.3–85.6) from 2014–2018. The individual survival rates of central nervous system tumors and lymphoma, two out of the three most common cancer types, have also increased over time. The curves of survival by time cohort in Figure 3 also depict an improvement in the region’s overall survival rates for childhood cancer.

Figure 4 compares the spatial distribution of cancer incidence (SIR) between Health Areas and Basic Health Zones. Compared to those of 2010 [41], in which the incidence was generally elevated in the northwest region of the region of Murcia, the maps show an elevated incidence in the southeast regions, particularly in Health Areas 1 and 8. In Figure 5, the SIR distribution over four different time periods demonstrates the spatial changes of incidence over time, from Health Area 4 in 1998-2003 to Health Areas 1 and 8 in 2016–2020. 

## 4. Discussion

This study depicts a compact program, integrating environmental and community health in CACS with a long-term follow-up of more than 20 years since 1998, which allows for the study of incidence and survival while also exploring new hypotheses in the etiology and prognosis of cancer in children ages 0 to 14 years in the region of Murcia, providing an update to regional data released over ten years ago [40]. Overall, there was an increase in the trend in the incidence of childhood cancer. The region’s crude rate had increased from 143.0 (per 1 million children) from 1998–2010 [40] to 149.6 from 1998–2020. However, the ASRw was lower between 1998–2020 than the rest of Spain (from 2000–2016) during a comparable time period. The region’s five-year survival rate (81.3%) was slightly higher than those observed in Spain (76% between 1995–2012) [40] and Europe (77.9% between 2000–2007) [51]. In France, the five-year survival rate was 80%. The survival program of France focuses on health problems (physical, psychological, social, and economic issues) that affect childhood cancer survivors and are needed to ensure that these patients regain the most optimal physical and emotional health possible [52]. However, it does not incorporate the environmental factors, ICF, community, and family participation in the process. In contrast, in Argentina, the five-year survival rate (2000–2009) was low, at 61% [53]. In less advantaged countries (Mexico, Peru, Ecuador), the survival rate for all leukemias (2010–2014) was from 32 to 46% [54]. The lack of trained personnel and access to tertiary health centers, limited treatment availability, cost of care, late stage of presentation, and treatment abandonment, contribute to the low survival rate [55].

Sex disparities in incidence and survival rates for CAC have been noted. In our study, the male-to-female ratio was 1:1, and females had a higher survival rate than males. These results align with the literature and may be caused by multiple factors such as sex differences in diagnosis delay, treatment response, birth weight, pharmacogenetics, tumor biology, or even treatment received [56,57]. A cross-sectional telephone survey conducted on childhood cancer survivors that included adolescent and young adult survivors of the pediatric departments of three regional hospitals in Hong Kong found that the majority of the male participants had rarely or never performed testicular self-examination [58]. 

By the second decade of their life, more than 60% of childhood cancer survivors will suffer from at least one chronic non-communicable disease related to the treatment they had received or associated environmental risk factors, or both [59]. Moreover, globally cancer survivors are eight times more likely than their siblings to experience many of these severe or life-threatening health conditions [8]. Childhood cancer survivors have a mortality rate 11 times higher than healthy controls [60,61]. Consequently, it is important to use psychosocial support, which includes psychological, academic, and social relationships, and to encourage self-care and self-advocacy in childhood cancer survivors.

With regard to the environmental risks, our study found that CACS living in environments with very poor air quality (high annoyance scores) had a lower survival rate. Although a causal link between outdoor air quality and childhood leukemia has been proposed [62], to our knowledge, this is the first work that associates outdoor air quality with the survival of CAC. Urban air pollution increases the risk of cardiovascular and respiratory infections that could contribute to increased morbimortality in patients with a vulnerable immune system [63]. Previous studies have shown that tobacco smoke exposure increases the mortality related to the treatment in children with leukemia [64]. Therefore, reducing urban air pollution could help improve the wellbeing and survival of CAC. However, more studies are necessary to consider the subtypes of cancer and to monitor air pollutants.

Furthermore, the changes in the spatial distribution in the region of Murcia of childhood cancer incidence should continue to be explored, given the clear shift from the northwest to the southeast (Health Area 4 to Health Areas 1 and 8) over 22 years. Increasingly numerous studies have shown how environmental exposures to pesticides, contaminated soils, tobacco, and globalizing lifestyles increase the risk of cancer and even worsen the prognosis [65,66,67,68,69]. These environmental factors, which often affect health and can be modifiable, should be increasingly highlighted during the patient’s intake process and throughout their medical journey. In the future, these data should be used to test for factors that may increase children’s risk of developing cancer. For this reason, we are analyzing this transition towards the coastal areas of the Cartagena Fields and Mar Menor fields in the southeast region of Murcia. It is essential to continuously develop and maintain a detailed environmental surveillance system and assess the multi-level effects of environmental factors on the appearance of childhood cancers in the region. This will also make it possible to continue using GIS to perform cluster analyses and explore spatial dependence in the distribution of cancer cases. The extensive information available can also be used for further multivariate analysis of childhood cancer incidence with associated variables such as family income, cancer type, and environmental exposures [70]. 

Consequently, it is important to incorporate worldwide the environmental factors that can prevent, promote, and mitigate childhood cancer. Additionally, the incorporation of the ICF could be used as an easily accessible and beneficial tool due to its ease of measure and its ability to promote environmental awareness, empowerment, and healthier lifestyles. Many environmental risk factors are shared among a number of non-communicable diseases. All recommendations for lifestyle and environment for CACS could contribute to attaining Sustainable Development Goals for non-communicable diseases within this significant population. Moreover, these healthy actions can translate directly to healthy children.

This analysis is part of the larger Long-Term Follow-Up Program for Childhood Cancer Survivors in the region of Murcia, which could help former patients retain their health for longer periods of time, focusing on preventing or managing late effects through surveillance and engaging a healthy environment and lifestyle for childhood cancer survivors in the region of Murcia. They operate under the Pediatric Environmental Health Specialty Unit (PEHSU) at the University Hospital of “Virgen de la Arrixaca”, working as a highly innovative initiative to integrate environmental assessment into clinical practice. This survival program gives an important role to the environmental factors and social determinants. Moreover, integrating simple metrics, such as ICF, could help improve monitoring and survival outcomes for CAC. A study monitoring repeated evaluations of ICF in the life span of the CACS could be a simple indicator of planetary health.

One asset pertaining to this study is the use of a long-term, extensive, and organized database that identifies the vast majority of children with cancer in the region of Murcia. It is unlikely that some cancer cases may have been missed, for instance, if they were to lack a diagnosis or have been misclassified. PEHis incorporates an individualized risk assessment that contributes to the development of new professional profiles and capacity building in environmental health. However, one limitation of this study would be a lack of multivariate analysis that would allow for the identification of causal relationships between factors of the environment and childhood cancer incidence and survival. Future studies should explore these scenarios and continue using GIS to visualize the spatiotemporal distribution of cases as they arise.

A cancer diagnosis is devastating at any age; however, it is especially devastating when said patient is a child. CACS are at greater risk of morbidity and mortality during their lifetimes due to late relapses, secondary tumors, and the early development of non-communicable diseases. Several studies have emphasized the need for lifestyle changes, such as reducing or eliminating environmental risks, as being among the most important goals in the long-term follow-up of CACS [40,59,71,72]. As many cancer-related complications do not become apparent until the survivor reaches adulthood, the implementation of timely interventions in Environmental Health could be critical in preventing or ameliorating late treatment sequelae and their adverse effects. After all, the social consequences of cancer are more deleterious for the most disadvantaged groups. For this reason, we are working on the development of an environment, survival, and childhood cancer (ENSUCHICA) program with an international and multidisciplinary collaborative consortium to improve the environmental health and the survivorship care of childhood cancer survivors through structured knowledge exchange, capacity building, and a global multidisciplinary collaborative approach [11].

## 5. Conclusions

After twenty years of experience, the PEHis could be an important tool for addressing childhood cancer. It captures an extensive array of clinical and environmental health data regarding childhood cancer survivors, for the goals of improving their long-term health, empowering survivors, and promoting the creation of healthier environments and lifestyles. Related to our study is improving the urban air quality that survivors breathe and supporting health information systems for their use in planning, management, and public health policies. 

The growing social awareness of the relationship between health and the environment is a major driver in achieving the UN’s sustainable-development goals. Gradually, more and more CACS express their concerns alongside their families, asking questions about the potential cancer risk factors, and how they can contribute to improving the prevention and the prognosis of the disease. The necessary empowerment of those affected by chronic and multifactor diseases, such as cancer, will help achieve these goals. We strive towards the possibility of getting to a global ten-year survival rate of 90% at the end of the decade, and we work to create a collaborative network with primary care. Incorporating the PEHis may play a major role in future in providing simple and engaging tools for self-monitoring patients; thus, it promotes environmental awareness, empowerment, and healthier lifestyles to safeguard planetary health. 

## Figures and Tables

**Figure 1 ijerph-20-00443-f001:**
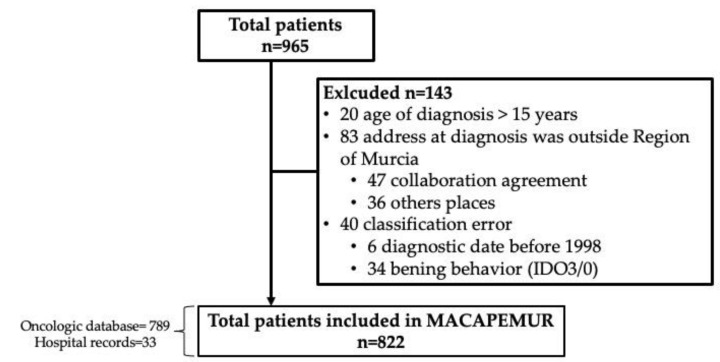
Total patients included in MACAPEMUR (1998–2020).

**Figure 2 ijerph-20-00443-f002:**
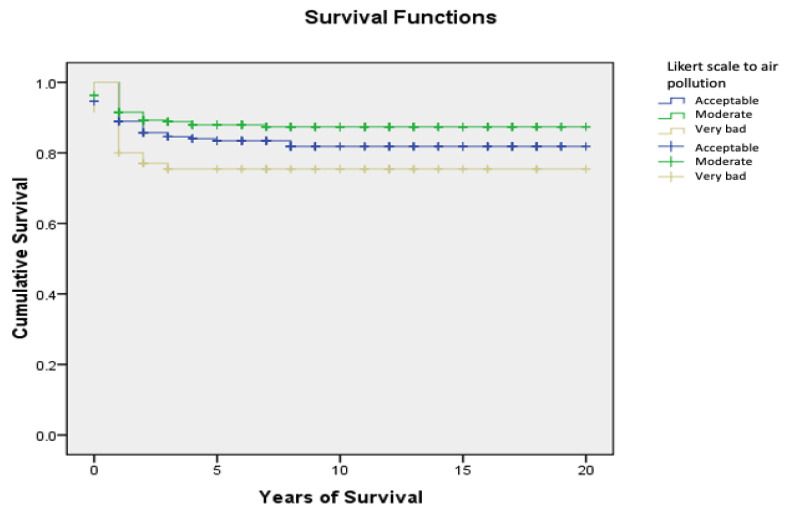
Cumulative survival per year of survival of the three groups of the Likert scale for air pollution in the region of Murcia (1998–2020).

**Figure 3 ijerph-20-00443-f003:**
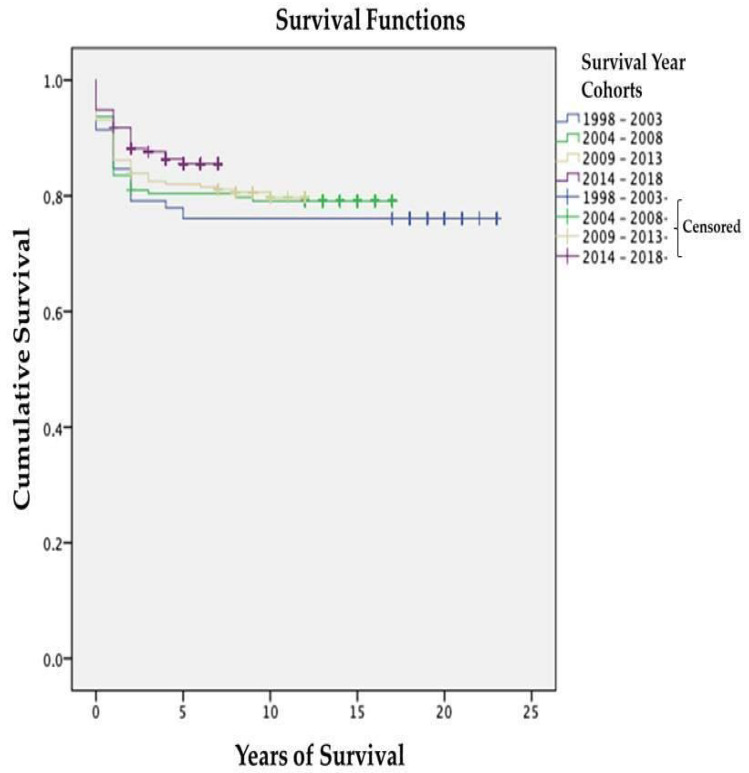
Childhood cancer survival across different time periods 1998–2018 (n = 733). The Cox proportional hazards regression model demonstrates that the survival rate for all childhood cancers has improved over time [log rank = 0.206].

**Figure 4 ijerph-20-00443-f004:**
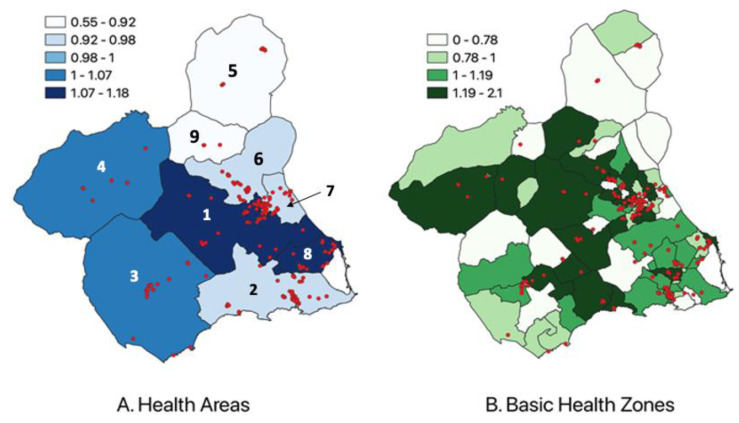
SIR by health areas and basic health zones in the region of Murcia, 1998–2020 (n = 822). Figure 4A presents the spatial distribution of the SIR for each of the 9 health areas in the region of Murcia. Figure 4B shows the spatial distribution of the SIR in the 90 basic health zones in the region of Murcia. The vector data correspond to the residential addresses of each case at the time of the cancer diagnosis. The maps are divided by the health areas and basic health zones that exist in 2021. SIR: standardized incidence ratio.

**Figure 5 ijerph-20-00443-f005:**
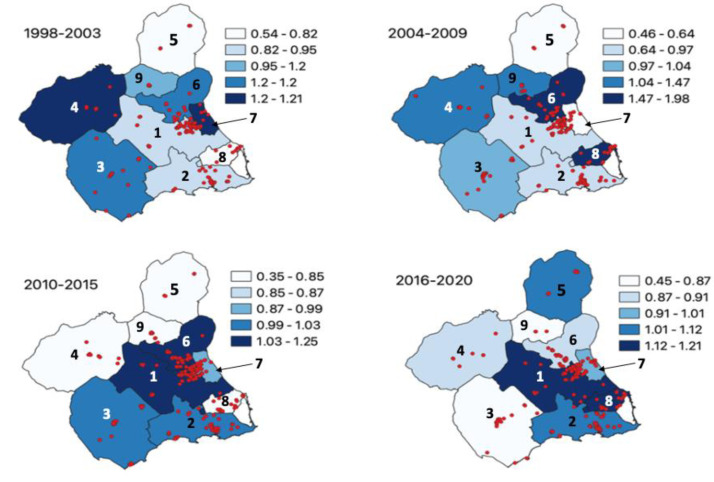
SIR by health areas and time cohort in the region of Murcia, 1998–2020 (n = 822).

**Table 1 ijerph-20-00443-t001:** Individual carbon footprint, Green Page, risk assessment, transition report, and individualized long-term follow-up plan.

Name	When?	Description
Individual Carbon Footprint	At diagnosis	o Individual carbon footprint (ton CO2e) measurement: - Questionnaire on personal activities that contribute to the carbon footprint.
Green Page	At the time of diagnosis	o Basic environmental screening, including: - Personal data. - Tumour classification (ICCC3). - Genealogical factors. - Geographical location (pregnancy, infancy, at the time of diagnosis). - Environmental tobacco-smoke exposure. - Chemical risk exposure in parents’ occupation. - Level contact of nature and air pollution perception. - Ambient air quality as measured by an air pollution annoyance scale. - Subjective etiological risk perception from parents.
Risk Assessment	1–4 months after diagnosis	o Careful environmental and genetic risk detection: - Descriptive questionnaire about risk factors described in the scientific literature as having any kind of association with the development of each type of tumour or the prognosis. - Evaluation of individual, familial, home, neighbourhood, and kindergarten/school risks. o Promotion of healthier lifestyles. o Psychosocial support (psychological, academic, social relationships). o Assessment and monitoring of health-related quality of life.
Transition Report “green passport”	At the end of the treatment	o Detailed report of the treatment: - Type of treatments and dosage. - Complications during treatment. o Late effects risk identification: - Individualized identification of potential late effects according to treatment. - Recommended follow-up. - Presence of any late effects or genetic condition. o Promotion of healthier lifestyles. o Environmental risk assessment associated with late effects. o Psychosocial support (psychological, academic, social relationships). o Assessment and monitoring of health-related quality of life.
Long-Term Follow-Up Plan	During the CACS’ lifespan	o Searching for opportunities for decentralized and shared care models between the ENSUCHICA Team (long-term follow-up unit) and primary care: - Screening for late effects. - Diagnosis and management of late effects. - Promotion of healthier lifestyles. - Environmental risk assessment associated with late effects. - Psychosocial support. - Assessment and monitoring of health-related quality of life. - Cause of death. o Encouraging self-care and self-advocacy in childhood and adolescent cancer survivors (CACSs).

**Table 2 ijerph-20-00443-t002:** Childhood cancer survival at 2, 5, and 10 years by distribution for the individual and socioeconomic characteristics and environmental exposure in the region of Murcia (1998–2020).

	% Survival (Typical Error)						
	2 Years		5 Years		10 Years		*p*-Value *
Sex							
Male	82.8	(1.8)	82.0	(1.9)	81.2	(1.9)	0.651
Female	86.3	(1.8)	83.3	(2.0)	81.6	(2.1)	
Income							
<2000	87.6	(1.6)	86.5	(1.6)	85.3	(1.8)	0.581
2000–3500	86.7	(2.8)	85.0	(3.0)	83.9	(3.1)	
>3500	90.9	(4.3)	-	-	-	-	
Family history of cancer 1st degree							
Yes	89.0	(4.7)	-	-	-	-	0.759
No	87.3	(1.3)	85.9	(1.4)	84.7	(1.5)	
Family history of cancer 1st degree or 2nd degree (younger than 55 years old)							
Yes	90.4	(2.8)	88.3	(3.1)	86.9	(3.3)	0.465
No	86.8	(1.5)	85.4	(1.5)	84.4	(1.6)	
Likert scale of air pollution in address at the time of diagnosis							
Acceptable	85.7	(2.5)	83.4	(2.7)	81.8	(2.9)	<0.05
Moderate	89.3	(1.8)	87.9	(2.0)	-	-	
Very bad	77.0	(5.0)	-	-	-	-	
Exposure to pesticides inside the residence							
Yes	86.9	(1.7)	85.7	(1.8)	84.4	(1.9)	0.57
No	88.0	(2.0)	86.6	(2.2)	-	-	
Contact with nature							
Every day	88.7	(1.8)	87.5	(1.9)	87.0	(2.0)	0.937
Once a week	88.6	(1.9)	86.7	(2.1)	85.2	(2.3)	
Once a month	87.9	(5.1)	-	-	-	-	
Only in vacations	-	-	-	-	-	-	
Never	-	-	-	-	-	-	
Physical activity frequency							
Never	90.0	(3.4)	88.5	(3.6)	-	-	0.283
1–2 days/week	90.1	(1.8)	88.9	(1.9)	-	-	
3–4 days/week	-	-	-	-	91.7	(2.7)	
> 5 days/week	96.3	(2.6)	-	-	-	-	
Maternal smoking during pregnancy							
Yes	88.1	(1.9)	-	-	85.2	(2.2)	0.824
No	86.8	(1.8)	85.7	(1.9)	-	-	
Paternal smoking during pregnancy							
Yes	86.1	(1.8)	84.8	(1.9)	83.4	(2.0)	0.231
No	89.0	(1.9)	88.2	(1.9)	-	-	
Maternal smoking at diagnosis							
Yes	85.9	(2.4)	85.3	(2.5)	83.5	(2.7)	0.352
No	88.9	(1.5)	87.5	(1.7)	-	-	
Paternal smoking at diagnosis							
Yes	85.9	(2.2)	85.0	(2.3)	83.6	(2.4)	0.327
No	88.5	(1.7)	87.1	(1.8)	-	-	

* Log rank (Mantel–Cox) was performed for the overall survival years with the 95th percentile.

**Table 3 ijerph-20-00443-t003:** Distribution of childhood cancer in the region of Murcia (1998–2020). Crude rate and standardized incidence ratio by tumor type and time cohort (n = 822).

		Cases by Period						CR/ASRw			SIR (CI 95%) by Type and Period			
		98–03	04–09	10–15	16–20	Total Periods	%	MACAPE MUR 98-20	Spain 00–16	S. Europe 01-10 ^a^	1998–2003	2004–2009	2010–2015	2016–2020
Leukemia		43	64	72	62	241	29.2	43.8	46.4/47.5	/51.1	0.81 (0.62–1.05)	1.01 (0.81–1.24)	1.04 (0.85–1.26)	1.12 (0.90–1.39)
	ALL	33	51	56	47	187		34.0	36.4/37.4	---	0.81 (0.59–1.08)	1.04 (0.81–1.31)	1.04 (0.82–1.30)	1.10 (0.85–1.40)
	AML	8	11	12	12	43		7.8	8.0/8.1	---	0.84 (0.42–1.51)	0.97 (0.55–1.61)	0.98 (0.56–1.58)	1.22 (0.70–1.97)
Lymphomas		21	12	33	19	85	10.3	15.4	20.2/19.7	/21.6	1.08 (0.73–1.56)	0.56 (0.32–0.91)	1.38 (1.01–1.85)	0.93 (0.61–1.37)
	HL	9	5	9	12	35		6.4	7.4/7.0	---	1.12 (0.59–1.08)	0.57 (0.81–1.31)	0.92 (0.82–1.30)	1.43 (0.85–1.40)
	NHL	12	7	24	7	50		9.1	6.8/6.7	---	1.06 (0.58–1.95)	0.56 (0.22–1.20)	1.70 (1.09–2.54)	0.59 (0.82–2.31)
CNST		41	47	65	42	195	23.7	35.4	35.8/36.1	/37.6	0.95 (0.72–1.23)	0.92 (0.71–1.18)	1.16 (0.94–1.43)	0.93 (0.71–1.21)
SNST		14	13	27	16	70	8.5	12.7	13.1/14.0	/14.5	0.92 (0.56–1.44)	0.68 (0.40–1.08)	1.33 (0.94–1.84)	1.04 (0.66–1.59)
Others		44	60	72	57	233	28.3	42.3	41.4/	---	---	---	---	---
**TOTAL**		163	196	269	196	824	100	149.6	157.0/159.4	---	0.89 (0.78–1.01)	0.91 (0.80–1.02)	1.14 (1.03–1.26)	1.03 (0.92–1.16)
CR		131.3	157.9	216.7	157.9	149.6								

Data adapted from Lancet Oncology ^a^. MACAPEMUR 98-20: Environment and Paediatric Cancer in the Region of Murcia from 1998 to 2020; CR: crude rate; ASRw: age-standardized incidence rate; SIR: standardized incidence ratio; ALL: acute lymphoblastic leukemia; AML: acute myeloid leukemia; HL: Hodgkin’s lymphoma; NHL: non-Hodgkin’s lymphoma; CNST: central nervous system tumors; SNST: sympathetic nervous system tumors.

**Table 4 ijerph-20-00443-t004:** Childhood cancer survival at 1, 3, 5, and 10 years after diagnosis by common tumor type and incidence during four time periods in the region of Murcia (1998–2018).

	Incidence Cohort	% Survival (Typical Error)			
		1 Year	3 Years	5 Years	10 Years
All Cancer (n = 733)	98–03	84.7 (2.8)	79.1 (3.2)	76.1 (3.3)	76.1 (3.3)
	04–08	83.5 (2.9)	80.4 (3.2)	80.4 (3.2)	79.1 (3.2)
	09–13	86.2 (2.3)	82.5 (2.6)	82.0 (2.6)	79.7 (2.8)
	14–18	91.8 (2.0)	87.6 (2.4)	85.5 (2.6)	---
	**98–18**	**86.8 (1.3)**	**82.7 (1.4)**	**81.3 (1.4)**	**80.1 (1.5)**
Leukemia (n = 207)	98–03	83.7 (5.6)	79.1 (6.2)	76.7 (6.4)	76.7 (6.4)
	04–08	92.2 (3.8)	92.2 (3.8)	92.2 (3.8)	92.2 (3.8)
	09–13	83.9 (4.7)	80.6 (5.0)	79.0 (5.2)	75.8 (5.4)
	14–18	94.1 (3.3)	90.2 (4.2)	90.2 (4.2)	---
	**98–18**	**88.4 (2.2)**	**85.5 (2.4)**	**84.5 (2.5)**	**83.3 (2.6)**
CNST (n = 79)	98–03	78.0 (6.5)	73.2 (6.9)	68.3 (7.3)	68.3 (7.3)
	04–08	71.1 (7.4)	71.1 (7.4)	71.1 (7.4)	68.4 (7.5)
	09–13	79.6 (5.5)	77.8 (5.7)	77.8 (5.7)	71.6 (6.9)
	14–18	87.0 (5.0)	78.2 (6.1)	78.2 (6.1)	---
	**98–18**	**78.0 (6.5)**	**73.2 (6.9)**	**68.3 (7.3)**	**68.3 (7.3)**
Lymphoma (n = 79)	98–03	81.0 (8.6)	81.0 (8.6)	81.0 (8.6)	81.0 (8.6)
	04–08	85.7 (13.2)	85.7 (13.2)	85.7 (13.2)	85.7 (13.2)
	09–13	91.3 (5.9)	91.3 (5.9)	91.3 (5.9)	91.3 (5.9)
	14–18	96.4 (3.5)	92.7 (5.0)	92.7 (5.0)	---
	**98–18**	**89.9 (3.4)**	**88.6 (3.6)**	**88.6 (3.6)**	**88.6 (3.6)**

CNST: central nervous system tumors. Bold: It is an overall data to the previous ones, it also separates the groups.

## Data Availability

The data presented in this study are available on request from the corresponding author. The data are not publicly available due to privacy.

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
