# Peer review of "Looking Towards 2030: Strengthening the Environmental Health in Childhood–Adolescent Cancer Survivor Programs"

_ijerph, 2022, doi:10.3390/ijerph20010443_

Round 1

Reviewer 1 Report

This is a study done using data from an impressive and comprehensive program to improve the health of young cancer survivors in the Region of Murcia, Spain especially by including a very strong environmental component.  It would be helpful if the end of the introduction provides a bit more information about why this particular study was done.  Most of the introduction provides a perspective of the importance of the environment on young cancer survivors’ health and well-being, but then a single sentence states that this study is going to look at incidence, survival and geospatial trends. I’m guessing that this is the first of potentially many papers tapping this rich research resource. Perhaps the authors undertook this particular study as a means for providing a basis for decisions about what factors to look at, directions to go more deeply into subsequently.  Let the reader know how this study fits into the program as a whole.

Abstract. Phrase “Increase in the trend of CC”. Increase in what – the incidence rate?

Section 2.1. Where is this program taking place? Who are the cancer survivors in the program?  Where do the patients come from (e.g., are they from a cancer registry, a children’s hospital)? How many of them are there? Are all of the patients in the program enrolled at time of diagnosis or do some come into the program later? What is the age range of patients?  Who responds to these questionnaires – the parents or the children (if they are old enough to)?  Are all children followed up or do some families drop out of the program?  Okay. Now that I have read to page 7, I see that a lot of this information is in section 2.3. Consider putting the second sentence in first paragraph of section 2.3 through the end of that paragraph earlier (perhaps in 2.1).  Section 2.3 could be re-labeled to something like “Descriptive and geospatial study.  Section 2.3 is missing a clear statement of the objectives – what is the scope of this study?

Graphic #6 is unreadable. The print is extremely small. It seems that this figure may be redundant with Table 1.

I read the description of the PCF at the bottom of page 5.  It is a “simple environmental health indicator that aims to reflect all the greenhouse gasses emitted indirectly by the cancer survivor.” How is it calculated?  What is included in it?

Page 6 Does “annoyance” mean the same as “disturbance” and how is this defined (e.g., by how it affects the person’s perceived well-being, by being bothered by odors or visibility issues due to smog, etc.?)  What is meant by “perceived risk of families”?

Page 8. Provide information about the distributions of duration of follow-up for the children in the study.

Page 8. “And their age was…” was this age at diagnosis?

Table 4. The text says the five-year survival rates are increasing over time for the three most common cancers, but this is not the case for leukemia.

Figure 5. What is the difference and meaning of “health zones” and “basic health areas”? Mention is made of Health areas 1, 7, and 8 but as far as I can tell there are no numbers on the areas in the maps.

Figure 6. What were the results from this figure (how are things changing over time and space)?  You mention this in the abstract and discussion, but don’t specifically state the findings in the results.

I think the conclusions in the Abstract go beyond the scope of the findings of the study.  While it is important to include the potential impact of the findings, the authors did not show that the integrating environmental health into clinical practices improves knowledge of the etiology, prognosis, and outcomes of childhood cancers.  However, the finding regarding pollution seems to be a potentially important clue about etiology. The authors did not report on the person carbon footprint indicator nor did they find out whether giving recommendations for lifestyle and environment promoted environmental awareness, empowerment, healthier lifestyles.  The conclusions should better reflect the implications of the findings of this particular study.  The program described in this paper appears to be a wonderful, forward looking and model approach that could lead to studies that more directly look at whether giving recommendations and incorporating environmental factors can lead to all the health benefits they list (empowerment, improved outcomes, etc.) for young cancer patients, but the findings in this paper don’t directly lead to those stated conclusions.

Minor comments

Page 2 Delete phrase “Due to their idiosyncrasies” – it is not clear and not needed.

Page 2 Phrase “children under fives years of age suffer from 40% of all environmental diseases” is not clear.

Page 2 Phrase “reference margins” is not clear.

Page 2 Phrase “difficult patients’ access” is not clear. Do you mean “decreases patients’ access”?

Page 2 ‘producing.  Is this the start of a quote?

Table 1 what does “Level contact of nature” mean?

Page 3 “the PCF, health-related quality of life and survival rate are global indicators of this program”. I am not sure what this means.  Are they “major indicators being evaluated by the program”? … Indicator of what?

Table 3. Provide a footnote that writes out wat MACAPMUR 98-20 is. Is LAL the same as ALL? LAM and AML? LH and HL and? LNH and NLH?

The page numbers do not seem to appear on all pages and restart with page 2 somewhere after page 10

Author Response

This is a study done using data from an impressive and comprehensive program to improve the health of young cancer survivors in the Region of Murcia, Spain especially by including a very strong environmental component.  It would be helpful if the end of the introduction provides a bit more information about why this particular study was done.  Most of the introduction provides a perspective of the importance of the environment on young cancer survivors’ health and well-being, but then a single sentence states that this study is going to look at incidence, survival and geospatial trends. I’m guessing that this is the first of potentially many papers tapping this rich research resource. Perhaps the authors undertook this particular study as a means for providing a basis for decisions about what factors to look at, directions to go more deeply into subsequently.  Let the reader know how this study fits into the program as a whole.

Response: Thank you for reviewing our manuscript. At the end of the introduction, we provided more information about why this particular study was done and how this study fits into the program as a whole. See changes in track changes.

Abstract. Phrase “Increase in the trend of CC”. Increase in what – the incidence rate?

Response: Thank you for the nice reminder. On the abstract, “Crude incidence rate” was added to the phrase “Increase in the trend of CC”. See changes in track changes.

Section 2.1. Where is this program taking place? Who are the cancer survivors in the program?  Where do the patients come from (e.g., are they from a cancer registry, a children’s hospital)? How many of them are there? Are all of the patients in the program enrolled at time of diagnosis or do some come into the program later? What is the age range of patients?  Who responds to these questionnaires – the parents or the children (if they are old enough to)?  Are all children followed up or do some families drop out of the program?  Okay. Now that I have read to page 7, I see that a lot of this information is in section 2.3. Consider putting the second sentence in first paragraph of section 2.3 through the end of that paragraph earlier (perhaps in 2.1).  Section 2.3 could be re-labeled to something like “Descriptive and geospatial study.  Section 2.3 is missing a clear statement of the objectives – what is the scope of this study?

Response: Thank you for reviewing our manuscript. Section 2.1 “Data collection” was added with the purpose of answering all the questions mentioned before presenting the tools and protocol of the program. Additionally, section 2.3 (now 2.4) was re-labeled to “Descriptive and geospatial Study” and scope was added. See changes in track changes.

Graphic #6 is unreadable. The print is extremely small. It seems that this figure may be redundant with Table 1.

Response: We understand your concern and decided to delete Graphic #6.

I read the description of the PCF at the bottom of page 5.  It is a “simple environmental health indicator that aims to reflect all the greenhouse gasses emitted indirectly by the cancer survivor.” How is it calculated?  What is included in it?

Response: We thank the Reviewer for having suggested this important point. We changed the name to individual carbon footprint (ICF) and added the definition in the introduction (page 2). We use the United Nations Carbon Offset Platform, which includes household, transport, and lifestyle information of the individual. This was included was added to page 5. See changes in track changes.

Page 6 Does “annoyance” mean the same as “disturbance” and how is this defined (e.g., by how it affects the person’s perceived well-being, by being bothered by odors or visibility issues due to smog, etc.?)  What is meant by “perceived risk of families”?

Response: Thank you very much for pointing this out. We added “disturbance” in brackets and the question we ask in the questionnaire (“How much does air pollution outside their home disturb them if they leave all the windows open”) on page 6 with the purpose of defining “annoyance” and what is meant by “perceived risk of families”. See changes in track changes.

Page 8. Provide information about the distributions of duration of follow-up for the children in the study.

Response: Thanks for your comment. The mean duration of follow-up for the children in the study was added on page 8. See changes in track changes.

Page 8. “And their age was…” was this age at diagnosis?

Response: Thank you very much for pointing this out. In the sentence “And their age was…” on page 8 was added, “…of this study”. See changes in track changes.

Table 4. The text says the five-year survival rates are increasing over time for the three most common cancers, but this is not the case for leukemia.

Response: Sorry for our inattention. This information was revised and corrected to “The survival rates of Central nervous system tumors and lymphoma, two out of the three most common cancer types, have also increased over time.” See changes in track changes.

Figure 5. What is the difference and meaning of “health zones” and “basic health areas”? Mention is made of Health areas 1, 7, and 8 but as far as I can tell there are no numbers on the areas in the maps.

Response: We thank the Reviewer for having suggested this important point. The difference and meaning of “health zones” and “basic health areas” was added on page 7. Numbers were added to the map on pages 13 and 14. See changes in track changes.

Figure 6. What were the results from this figure (how are things changing over time and space)?  You mention this in the abstract and discussion, but don’t specifically state the findings in the results.

Response: Sorry for our inattention. The findings were added to the results on page 13. See changes in track changes.

I think the conclusions in the Abstract go beyond the scope of the findings of the study.  While it is important to include the potential impact of the findings, the authors did not show that the integrating environmental health into clinical practices improves knowledge of the etiology, prognosis, and outcomes of childhood cancers.  However, the finding regarding pollution seems to be a potentially important clue about etiology. The authors did not report on the person carbon footprint indicator nor did they find out whether giving recommendations for lifestyle and environment promoted environmental awareness, empowerment, healthier lifestyles.  The conclusions should better reflect the implications of the findings of this particular study.  The program described in this paper appears to be a wonderful, forward looking and model approach that could lead to studies that more directly look at whether giving recommendations and incorporating environmental factors can lead to all the health benefits they list (empowerment, improved outcomes, etc.) for young cancer patients, but the findings in this paper don’t directly lead to those stated conclusions.

 Response: We understand your concern and have done our best to revise the mistakes. We agree that both the abstract and conclusion language misinterpreted our findings. For that reason, the conclusions in the two sections, abstract and conclusion, were reviewed and changed to more adequate language regarding our findings. See changes in track changes.

Minor comments

Page 2 Delete phrase “Due to their idiosyncrasies” – it is not clear and not needed.

Response: Revised accordingly. The phrase “Due to their idiosyncrasies” was deleted as recommended. See changes in track changes.

Page 2 Phrase “children under fives years of age suffer from 40% of all environmental diseases” is not clear.

Response: Thank you very much for pointing this out. On page 2, the phrase “children under fives years of age suffer from 40% of all environmental diseases” was deleted. See changes in track changes.

Page 2 Phrase “reference margins” is not clear.

Response: Thank you for these observations. On page 2, the phrase “reference margins” was deleted. See changes in track changes.

Page 2 Phrase “difficult patients’ access” is not clear. Do you mean “decreases patients’ access”?

Response: Thank you for these observations. On page 2, the phrase was changed to “disrupts access …”. See changes in track changes.

Page 2 ‘producing.  Is this the start of a quote?

Response: Thank you for these observations. On page 2, “‘producing” was deleted. See changes in track changes.

Table 1 what does “Level contact of nature” mean?

Response: Thank you very much for pointing this out. The level of Contact with nature is the frequency they do activities in contact with nature (e.g., going to neighborhood parks, orchards, mountains, beaches, and forests). On table 1, the meaning of “Level contact of nature” was added in the last sentence on page 5 and at the beginning of page 6.  See changes in track changes.

Page 3 “the PCF, health-related quality of life and survival rate are global indicators of this program”. I am not sure what this means.  Are they “major indicators being evaluated by the program”? … Indicator of what?

Response: Thank you for these observations. On page 3, the sentence was changed by adding “evaluated” and added what they are indicators of.

Table 3. Provide a footnote that writes out wat MACAPMUR 98-20 is. Is LAL the same as ALL? LAM and AML? LH and HL and? LNH and NLH?

Response: Sorry for our inattention. On Table 3, LAL was changed to ALL, LAM was changed to AML, LH was changed to HL, and LNH was changed to NLH, as recommended. See changes in track changes.

The page numbers do not seem to appear on all pages and restart with page 2 somewhere after page 10

Response: Sorry for our inattention. The pages were corrected and now appear on all pages as recommended.

Reviewer 2 Report

see attached file

Author Response

A very meaningful and highly valued project for the health maintenance and promotion of cancer survivors.

Response: Thank you very much.

  1. While the title is facilitating, it does not reflect the main design and findings of this study. I would suggest a revision of the title to reflect the current study.

Response: While we appreciate the reviewer’s feedback, we respectfully disagree. “Looking towards 2030: strengthening the Environmental Health in Childhood-Adolescent Cancer Survivor Programs”. We think the title reflects the twenty years of work of our program, which contains long-term (1998-2020) and extensive information about the pediatric environmental history of CACS in the Region of Murcia, Spain. We understand that our program stills need more studies to reflect how our program could improve the well-being and healthy environmental lifestyle of CACS in the Region of Murcia, Spain. For that reason, we are looking towards 2030 to achieve the UN's sustainable-development goals.

  1. The authors define personal carbon footprint (PCF) on p.5. I would suggest putting this definition earlier in introduction to facilitate readings. May also give some examples of personal activities that contribute to PCF, showing some of these activities are modifiable.

Response: We thank the Reviewer for having suggested this important point. We changed the name to individual carbon footprint (ICF) and added the definition in the introduction (page 2), and some examples of how to reduce our ICF. For the ICF, we use the United Nations Carbon Offset Platform, which includes household, transport, and lifestyle information of the individual. This was included on page 5. See changes in track changes.

  1. I suggest the authors take reference to the one relevant paper:

Health behaviors of Chinese childhood cancer survivors: A comparison study with their siblings. (2021). International Journal of Environmental Research and Public Health, 17(7):6136.

Response: We thank the Reviewer for having raised this important point. And decided to add this relevant paper as recommended. We reference this relevant paper in the discussion on page 14. See changes in track changes.

  1. The paper introduces a lot of variables , tools, materials, while the finding does not seem

to cover of them. I would suggest adding research objectives, and describe what part of a tool achieve what objective. Result and discussion can be presented following the sequence of the objectives.

Response: Thank you very much for pointing this out. In the introduction (page 3) and abstract, the aims of the study were revised, and the section “Process” from the Materials and Methods was re-labeled to “Descriptive and Geospatial Study” and mentioned that, for the analysis, this study only included information recollected from the “HverdeGEO” and the clinical history of the CACS. See changes in track changes.

  1. I suggest adding a section of data collection. It is unclear when the data is collected, how the data are collected (by self-report, observation, or from medical record or other big data base)?

Response: We agree with the reviewer and decided to add a section on Data Collection on page 3, as the section that initiates “Material and Methods”. See changes in track changes.

  1. Is the project ongoing or stop at a particular year?

Response: Thank you very much for pointing this out. This project is an ongoing project. For that reason, we added in the abstract “1998 to date” and the introduction (page 3) that is still ongoing to clarify any doubt. See changes in track changes.

  1. I can see there are some intervention in the program, e.g. psychological support, life style

promotion. How did these interventions affect the findings? Would the findings be

worse without these interventions. It’s worthwhile to add some discussion on this aspect.

Response: Thank you very much for pointing this out. On page 15, in the discussion, we added how important these interventions are to CACS.

  1. The paper has too many abbreviations. May consider using more full terminologies to

ease reading.

Response: Thank you very much for pointing this out. We reduced the abbreviations by deleting 6 and using more full terminologies to ease reading as recommended.

Overall, a very comprehensive and valuable project.

Response: Thank you for reviewing our manuscript.

Reviewer 3 Report

The authors have reviewed twenty years of experience in childhood cancer survivor programs. They have done an excellent job in this review. The PEHis will continue to be an important tool for addressing CC in the RM. It captures an extensive array of clinical and environmental information about children survivors with cancer for both goals of improving their long-term health, empowering survivors, and promoting the creation of healthier environments and lifestyles. Related with our study, improving the urban air quality that survivors breathe and supporting health information systems for their use in planning, management, and public health policies.

The growing social awareness of the relationship between health and the environment is a major driver to achieve the UN's sustainable-development goals. Gradually, CACSs and their families express their concerns; ask questions about the potential cancer risk factors, and how they can contribute to improving the prevention and the prognosis of the disease. The necessary empowerment of those affected by chronic and multifactor diseases, such as cancer, will help achieve these goals. We work with the aim to get to a Global ten-year survival rate of 90% at the final of the decade. We work to create a collaborative Network with Primary Care. Incorporating the PEHis and PCF will play a major role, such as simple and engaging tools for patient self- monitoring; thus, it promotes environmental awareness, empowerment, and healthier lifestyles to safeguard the Planetary Health."

Author Response

The authors have reviewed twenty years of experience in childhood cancer survivor programs. They have done an excellent job in this review. The PEHis will continue to be an important tool for addressing CC in the RM. It captures an extensive array of clinical and environmental information about children survivors with cancer for both goals of improving their long-term health, empowering survivors, and promoting the creation of healthier environments and lifestyles. Related with our study, improving the urban air quality that survivors breathe and supporting health information systems for their use in planning, management, and public health policies.  The growing social awareness of the relationship between health and the environment is a major driver to achieve the UN's sustainable-development goals. Gradually, CACSs and their families express their concerns; ask questions about the potential cancer risk factors, and how they can contribute to improving the prevention and the prognosis of the disease. The necessary empowerment of those affected by chronic and multifactor diseases, such as cancer, will help achieve these goals. We work with the aim to get to a Global ten-year survival rate of 90% at the final of the decade. We work to create a collaborative Network with Primary Care. Incorporating the PEHis and PCF will play a major role, such as simple and engaging tools for patient self- monitoring; thus, it promotes environmental awareness, empowerment, and healthier lifestyles to safeguard the Planetary Health."

Response: Thank you for reviewing our manuscript. The conclusion was changed to “After twenty years of experience, the PEHis could potentially be an important tool for addressing childhood cancer. It captures an extensive array of clinical and environmental health data regarding childhood cancer survivors, for both goals of improving their long-term health, empowering survivors, and promoting the creation of healthier environments and lifestyles.  Related to our study, improving the urban air quality that survivors breathe and supporting health information systems for their use in planning, management, and public health policies. The growing social awareness of the relationship between health and the environment is a major driver in achieving the UN's sustainable-development goals. Gradually, more and more CACSs express their concerns alongside their families; asking questions about the potential cancer risk factors, and how they can contribute to improving the prevention and the prognosis of the disease. The necessary empowerment of those affected by chronic and multifactor diseases, such as cancer, will help achieve these goals. We strive towards the possibility of getting to a Global ten-year survival rate of 90% at the final of the decade, and we work to create a collaborative Network with Primary Care. Incorporating the PEHis may play a major role in the future, providing simple and engaging tools for self-monitoring patients; thus promoting environmental awareness, empowerment, and healthier lifestyles to safeguard the Planetary Health. “

Round 2

Reviewer 1 Report

The authors did a great job of addressing my comments and concerns.  

Two minor comments are:

In the abstract re: "An increase in the trend of incidence of childhood cancer was found with a crude incidence rate of 149.6 cases per 1 million." It might be clearer to say. While the crude incidence rate across that entire period was 149.6 per 1 million, there was an increase over that time in the incidence..."  

In the intro re: "It is situated between the range of the incidence of pediatric cancer in the European area, which oscillates between 130 and 160 cases per million children under 15 years of age ".  It might be clearer to say: "It is at the lower end of the range of incidence in pediatric cancers across European countries of 130 to 160 cases per million children under 15 years of age".  

Author Response

The authors did a great job of addressing my comments and concerns.  

Response: Thank you very much for reviewing our manuscript.

Two minor comments are:

In the abstract re: "An increase in the trend of incidence of childhood cancer was found with a crude incidence rate of 149.6 cases per 1 million." It might be clearer to say. While the crude incidence rate across that entire period was 149.6 per 1 million, there was an increase over that time in the incidence..."  

Response: Thank you for reviewing our manuscript. The phrase was changed according to the suggestion. See track changes.

In the intro re: "It is situated between the range of the incidence of pediatric cancer in the European area, which oscillates between 130 and 160 cases per million children under 15 years of age ".  It might be clearer to say: "It is at the lower end of the range of incidence in pediatric cancers across European countries of 130 to 160 cases per million children under 15 years of age". 

Response: Thank you for reviewing our manuscript. The phrase was changed according to the suggestion. See track changes.
